# Emerging Patterns of Mountain Tourism in a Dynamic Landscape: Insights from Kamikochi Valley in Japan

**Abhik Chakraborty**

Faculty of Tourism, Wakayama University, Wakayama 640-8510, Japan; abhich78@wakayama-u.ac.jp or abhichkro@gmail.com; Tel.: +81-73-457-8564

**Abstract:** This article analyzes the emerging contours of mountain tourism in a highly popular destination in the North Japan Alps by reporting the findings of a two-year long study at the Kamikochi Valley. The main aim was to understand the dynamic character of the biophysical landscape and the perceptions of tourism service providers and visitors. The study was conducted using a qualitative design and involved in-depth interviews, observations, and a questionnaire survey for visitors. It was found that while different stakeholders held different perceptions of the landscape, there was a general lack of understanding among tourism service providers and visitors regarding the relationship between long-term processes and fine-scale heterogeneity of the landscape. The prevalence of an engineering approach has led to sweeping changes of key landscape interaction pathways over the years, threatening the heterogeneity and resilience of the natural environment. The findings also indicate a general visitor demand of information on the biophysical environment, and therefore it is of urgent need to address the biophysical integrity of such landscapes, and raise visitor awareness through the provision of relevant information.

**Keywords:** mountain destination; dynamic landscape; heterogeneity; geological time; anthropogenic modification; North Japan Alps

## 1. Introduction

This article presents the outcomes of a two-year long research project at the Kamikochi Valley of North Japan Alps that assessed sustainable tourism challenges from a landscape point of view. Mountains occupy an important position in the international tourism landscape: collectively they attract 15–20% of global tourists, making them the second most popular destination category after islands and beaches [1]. Mountain environments are typically dynamic due to their physical properties and processes such as high relief, seasonal variations, and denudation and transport regimes [2–4]. Vigorous physical regimes also imply that mountain landscapes are more than passive backdrops to human activities of meaning-making and constructing landscapes, the intractable materiality of mountains interact actively with human schemes [5,6], and frequently pose difficult questions regarding satisfactory management of such places [7]. In addition, the history of land use is an important factor influencing landscape characteristics of mountain destinations: mountains have been inhabited or used for millennia by local societies [8–10]. While long-term human interaction with mountains can also engender landscape heterogeneity and maintain socio-ecological landscapes over time [8,9], an overall trajectory of clearing of original vegetation and intensification of impact during modern times has been observed [8,10]. Some recent works variously contend that mountain landscapes are also vulnerable under accelerating global environmental change [4], that mountain destination management in different countries have different priorities and perceptions [11], and that that summer visitation will further intensify under a warming climate, but mountain destinations are inadequately prepared for such change [12]. Regarding the issue of managing change in mountain destinations,

a number of works are available on the topic of climate change [13–15] as well as cultural constructions of mountain landscapes [16,17], but relatively little literature is available on anthropogenic alteration of physical processes that engender landscape diversity of mountain destinations. This is a major research gap and a pertinent point of inquiry, as it has been observed that humans currently influence major geomorphological processes in mountain regions, resulting in a far-reaching effect on the integrity of those landscapes [18–20]. As tourism in mountains is highly context specific [1], and as visitors typically have a complex range of preferences and needs [21–23], case studies that offer insights on the specificities of mountain destinations and challenges are clearly of much relevance and import.

This study focused on the Kamikochi Valley of North Japan Alps, which is one of the most popular mountain destinations in the Japanese Islands, in order to analyze recent anthropogenic changes in landscape characteristics and perceptions of tourism stakeholders and visitors. It was a part of a four-year long larger research project on the North Japan Alps area that is currently ongoing. The principal aims of this study were to (i) highlight the dynamic properties of this landscape and (ii) describe the perception of service providers and visitors. As there is a scarcity of research literature in English available on the Japan Alps, this case study makes a timely, important, and clear contribution to the field.

## 2. Description of the Study Site

Kamikochi is a valley located within the Azusa River watershed in the North Japan Alps (Figure 1). The Azusa River finds its headwaters in the highest peaks of the North Japan Alps—Mt. Yari (3180 m. asl.) and Mt. Hotaka (3190 m. asl.)—and flows down by the Matsumoto basin to eventually form the longest river of Japan, the Shinano (370 km). The upper parts of the watershed are known for large glacier eroded valleys such as the Yarisawa—and are also home to some of the most vigorous uplift and denudation processes in the Japanese Islands [24]. While uplift and crustal deformation remain principal drivers of elevation, recent research has demonstrated that the peaks of Mt. Yari and Mt. Hotaka are remnants of a large Quaternary caldera volcano [25,26]; and Mt. Yake (2455 m. asl.), an active volcano, still spews smoke by the riverside.

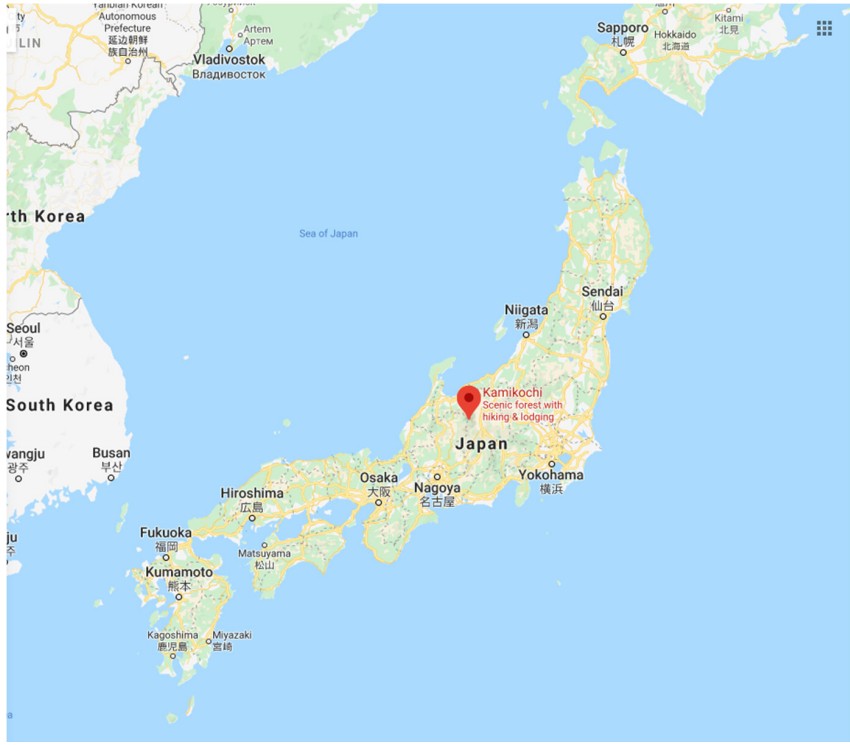

**Figure 1.** Location of Kamikochi in Japan (Google Maps).

Kamikochi is a major gateway for the peaks of the North Japan Alps. While accurate visitor statistics are not available (a common problem even for major visitor destinations in Japan) a total 'usage frequency' of 1.27 million was recorded in 2014 [27]. Although this number is potentially inflated by multiple use of the same facility by a single user, the figure still indicates significant visitor pressure on this landscape, resulting in ongoing tension between visitor demand for an enjoyable environment and the inherent dynamism of the mountain landscape [28].

The biophysical landscape of Kamikochi is characterized by active riverbed formation of the Azusa River (Figure 2), which is further driven by uplift, deformation, and denudation processes operating at the highest ridgeline of North Japan for over 1.7 million years [24,29]. Recent glaciation events left their imprints in the form of large valleys scoured by glacial erosion, through which the headwaters of the river flow. These dynamic processes cause frequent landslides, movement of boulders and coarse gravel on the riverbed, and flooding. During the past 100 years, human modification of this landscape has intensified. Major modification of the basin hydrology began with river engineering and road construction in the late 19th century. In the 20th century construction of recreation and accommodation facilities followed, and further channel modification took place. The relationship between dynamic physical agents of the landscape operating over millions of years and human agents modifying it for a few hundred years therefore encapsulates a constant tension.

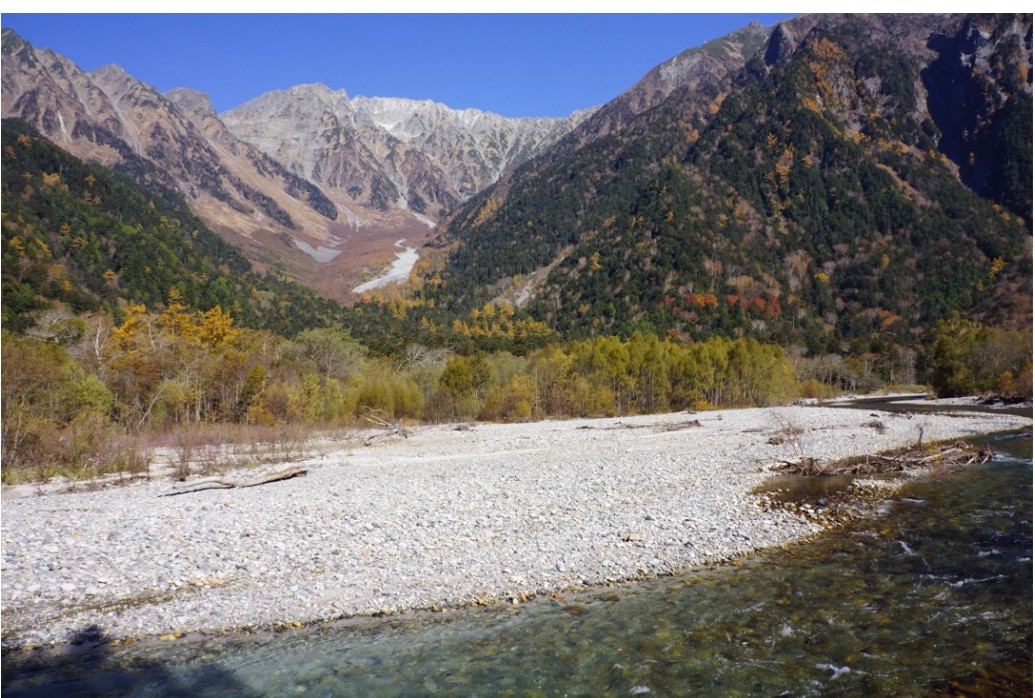

**Figure 2.** Kamikochi, with Azusa River in the foreground, *Salix arbutifolia* growing on the gravelly riverbed, and the Hotaka Range in the background. Photo by author.

Before Japan's modernization, Kamikochi was only accessible by a 44 km long trek through the Tokugo Pass from Matsumoto. Available records indicate that prior to Japan's European-style modernization in late 19th century, villagers working under the Matsumoto fief logged local forests for several centuries; timber extracted from the forest was floated down the Azusa River to Matsumoto [30]. As a result of such activities, substantial sections of the mountain forest were logged off during the Edo Period (1603–1868). Murakushi (2005) [31] detailed the transformation of the valley during the premodern, early modern, and post-World War II periods. During the early modernization of Japan at the end of the 19th century, consolidation of the central state led to restrictions on logging and the forests recovered somewhat, even as road and hydroelectric engineering began. Side by side, landscape modification by planting Japanese larch (*Larix kaempferi*) around what is the present-day bus

terminal, ensued [30]. Starting from 1885, there was a brief period when pasturing was introduced to the area, but as tourist use of the landscape became increasingly popular in the 20th century, pasturing moved to the background and was eventually phased out in 1934. Early form of mass tourism is dated at 1909 [31], but the major phase of touristic development in the valley began with the opening up of the Kama Tunnel in 1933 [32]. The Kama Tunnel forms the main artery of transport to this day. The Kamikochi Bus Terminal is located at 1500 m. asl. and is used by most visitors to this area. Thus, an overall pattern of incremental impact of recreational use on the landscape throughout the 20th century can be discerned.

This picturesque valley also played a crucial role in the formulation of Japan's National Park system. In the early 20th century, a plan was mooted to construct a large dam that would have submerged the entire valley under an artificial reservoir. Tsuyoshi Tamura and Seiroku Honda, influential figures who shaped the early National Park movement, opposed the scheme and emphasized tourism as an alternative development pathway for Kamikochi [31,32]. Kamikochi was successfully protected when it was registered inside the Chubu Sangaku National Park in 1934. The Chubu Sangaku National Park is one of the most important National Parks in Japan, and is among the largest national parks in the Honshu Island. After the opening of the Kama Tunnel, a rapid increase of visitors ensued in the mid-1950s, and the popularity of the valley also increased due to its portrayal in the novel 'Hyoheki' (Ice-wall) by the famous novelist Yasuhi Inoue [33]. Increased tourism in turn created the problems of littering, air pollution, and traffic congestion during the middle of the 20th century, before private car access was eventually blocked in 1975 [33]. Today, although the valley retains its attractive scenery, a number of dams just below Kamikochi have rendered the flow of the Azusa River largely artificial and there is ongoing modification of the river even within the National Park area [29,34]. In addition, a proliferation of roads, numerous trails, and accommodation facilities have contributed to the steady increase of human footprint in the valley.

## 3. Materials and Methods

The main findings are based on three components: content analysis of a document that reports long-term monitoring of the place; information gained from 7 in-depth (open-ended) interviews with local stakeholders and personal observations of the author; and data from a sample of 80 valid questionnaires (Figure A1) aimed at visitors.

During the research, a qualitative case study method was followed [35–37]. The case study was a part of a four-year long ongoing research project on the North Japan Alps area. The spatial unit of the Kamikochi valley was chosen as a 'case', in an approach in consistence with Swanborn (2010). [38] The case was selected because of its intrinsic importance [35]: as described above, Kamikochi played a crucial role in the formulation of Japan's National Park system in the early 20th century [31], and it remains one of the two most prominent gateway locations in North Japan Alps [28]. The case was also chosen for its instrumental importance [35] as an instructive example for highly visited mountain landscapes. Due to the lack of any systematic study on visitor or tourism stakeholder perceptions in this area, the research had to adopt an exploratory approach; i.e., it did not aim to analyze causality between already identified variables; instead, the aim was to describe the case and identify possible points for future management input. The research spanned a period of nearly two and half years—from April 2017 to October 2019—during which the spring-to-autumn season (April to October) was mainly utilized for data gathering (due to the fact that the area is closed during winter and early spring). A combination of data collection techniques—observation of the landscape, content analysis, photography, open-ended interviews with tourism service providers and national park management, and a structured questionnaire survey—was used to collect data, in consistence with standard qualitative data collection procedures [39–42]. A major source of data for understanding anthropogenic change to the landscape properties was a compilation by a group of local conservation scientists who have conducted research on landscape conservation through multiple years. This account, Natural History in the Kamikochi Valley [43] remains, to the knowledge of this

author, the most accurate and substantial account of the changes in the natural landscape of Kamikochi. In addition, a number of scientists who took part in the compilation and the National Park staff were approached for follow up questions and interviews. For the structured questionnaire survey, an initial trial (pilot phase) was conducted between June and October 2018 in order to gauge responses and improve the design of the questionnaire. Subsequently, the questionnaire was refined and formally implemented during June to October 2019. Due to the fact that most hikers were tired or in a hurry, and could not spend more than 2–3 min to fill out responses, and also due to the fact that most Japanese hikers are not accustomed to take part in surveys, the questionnaire had to be simple. It consisted of multiple choice type questions and columns to indicate the gender and age of the respondent. Due to local constraints (not all facilities would agree to implement the survey, and there were insufficient provisions for running the survey and storing data in other locations) a mountain hut was chosen to administer the survey. The facility—Tokusawa-en (Figure 3)—has a long history of nearly a hundred years, and is highly popular among hikers. Larger hotels that are located at the outskirts of the valley were not selected as hikers rarely choose them for lodging, and most of them are located outside the main study site. Besides, the manager and the staff of Tokusawa-en were cooperative and followed the instructions for data gathering accurately, which solved the problem of running the survey in an incorrect manner. A total of 200 questionnaires were distributed out of which a total of 80 completed samples were collected—i.e., the turnover rate was 40%. Open-ended interview data were analyzed through standard qualitative techniques such as coding and identification of key themes [39,44] and descriptive statistics was used for reporting the survey findings.

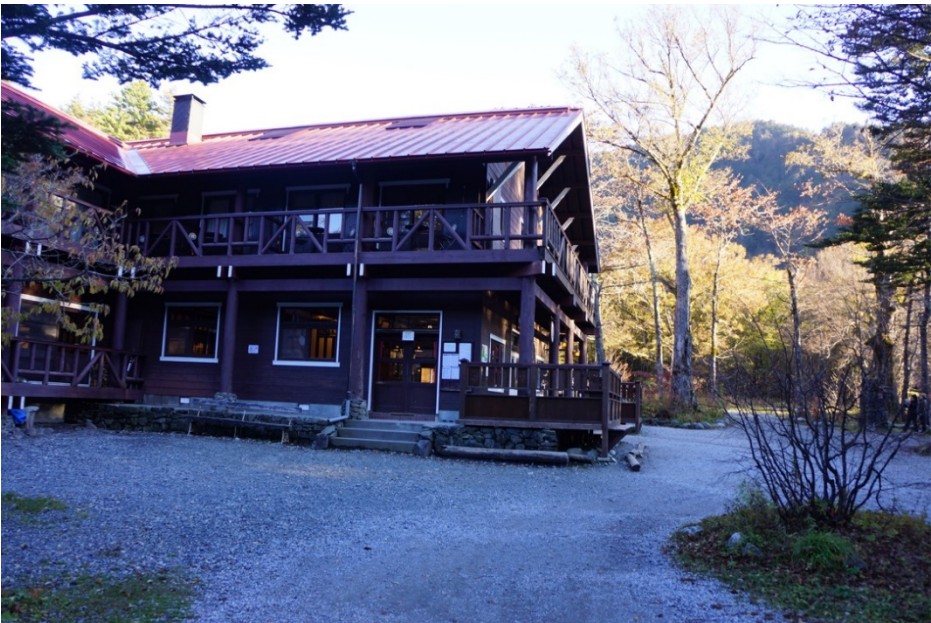

**Figure 3.** Tokusawa-en Mountain Hut. Photo by author.

## 4. Results

In this section, findings are reported in three sub-sections: (i) content analysis from long-term monitoring of the place by conservation scientists, (ii) 7 in-depth interviews and personal observations of the author, and (iii) structured questionnaire survey that yielded 80 valid responses. The sub-sections therefore also conform to the actual chronological sequence of the research project: research and analyses pertaining to (i) and (ii) were conducted during April 2017 to March 2018, and research pertaining to (iii) was conducted between June 2019 and October 2019 and the data were analyzed subsequently.

*4.1. Results of Content Analysis (Secondary Data): Characteristics of Active Landscape Formation in Kamikochi Valley*

As noted above, a detailed account of the natural environment of the Kamikochi Valley was recently compiled by a group of local conservation scientists who studied the place for nearly three decades [43]. This compilation is especially valuable, as it provides insights from long-term monitoring of the environment—a rarity in environmental research literature in Japan. The principal characteristics of this dynamic landscape as documented in this work and pertinent information from more general literature are summed up below as main results of content analysis:

Kamikochi is a landscape shaped by intense tectonic uplift, Quaternary volcanism, and glaciation. The overall mechanism of uplift and denudation is illustrated by Iwata (2016) [24]. A long rocky ridgeline joining the two peaks of Mt. Yari (3180 m.) and Mt. Oku-Hotaka (3190 m.) forms the main chain of mountains. Although they were formerly thought of as being uplift-induced, these peaks were later ascribed a volcanic origin [25] dating 1.76 million years ago. The peaks are likely to have formed due to a complex combination of explosive caldera volcanism, magma induced uplift, and subsequent erosion related enhancement of relief.

The Azusa River, the main feature of the Kamikochi Valley, is known to have changed course in geological time in response to volcanic deposition and land formation [24,45]. The wide valley of Kamikochi is somewhat counterintuitive as it sits upstream of a narrow gorge-like section of the river; it is conjectured that a significant phase of volcanic activity of the Mt. Yake volcanic group that began 26 Kya might have blocked off the riverflow and formed a large lake (~16 Kya), which subsequently drained away, leaving the cavity open to be filled up with deposition from ridgeline erosion, mass movement, and transport by the river. In addition, lava flow of more recent origin (~4 Kya) blocked off sections of the river donstream from Kamikochi, resulting in the gorge-like landscape formation downstream [45]. The multi-thread channel in the Kamikochi Valley—where the river flows in several streams on a wide gravelly bed—is induced by a complex range of factors such as past volcanism, Quaternary ridge formation, subsequent glacial erosion, as well as vigorous mass movement/denudation in the Holocene.

However, this complex evolution of the landscape in geological time is not adequately perceived at the planning level. The Taisho Pond, which was formed in 1915 when lava flowing out of Mt Yake blocked off the river flow, is artificially kept alive to appease tourist interest by a concrete weir that blocks the natural mechanism of the river to drain the small lake [29]. Small rocky tributaries that are vital conduits of transport in the watershed are blocked off or altered by small-scale weir construction and embankment engineering [34]. In particular, tourism related infrastructure buildup has had the effect of constraining the propensity of the river to flow in a multi-thread channel and limiting its floodplain dynamics, as well as impacting fine-scale heterogeneity of tributary streams. Several hotels and accommodations are currently located within the historical floodplain of the Azusa River [29].

Several species are possibly impacted due to anthropogenic modification of natural regimes, with the Chosenia (Salix arbutifolia) vegetation frequently cited as an indicator case [34,46,47]. Once found widely in Honshu, these riverbed vegetation colonies largely disappeared during the 20th century as rivers were modified in extensive scale all over Japan. Being a pioneer species, S. arbutifolia thrives on periodic disturbances such as flooding and in-channel gravel deposition. Kamikochi currently forms the last large-scale natural habitat for the species in the Honshu Island, but the future of the species is under threat in Kamikochi due to the suppression of natural disturbance regimes of the Azusa River. Iwata and Yamamoto (2016) [34] observed instances of flood intolerant species like Ulmus davidiana and Abies homolepis expanding their ranges in riparian sections that were formerly dominated by S. arbutifolia, but were subsequently subjected to flood controlling mechanisms.

*4.2. Results of Interviews and Observations*

During the first year of the project, interviews with local tourism service providers, National Park staff, and conservation scientists were conducted in order to understand the main contours of

tourism in the valley and management of its dynamic landscape. These interviews were open ended, and ranged from casual conversations to hour-long discussions. A total of 7 in-depth interviews each spanning nearly an hour were the main sources of relevant information. The information derived is summed up below:

Category I. Service providers (Mountain huts): Typically, interviewees who worked in the mountain huts sought to portray Kamikochi as the perfect escapade for busy urban customers. Key words used by them were: a gentle place which does not require climbing skills or long hiking endurance in order to visit, soothing shade, water, and spectacular views of the North Japan Alps Range, and provision of relaxation. The manager of Tokusawa-en, a man in his 40s, proudly pointed out that his was the oldest accommodation facility in Kamikochi, and the popularity of the hut among hikers also partially stemmed from the fact that the famous novelist Yasushi Inoue mentioned it in his novel 'Hyoheki'—which incidentally became a major cause behind Kamikochi's popularity in postwar Japan, as described before. Accordingly, the hut sought to maintain its identity as a retreat for literary or artistic minded customers—many of its current lodgers are said to be painters and photographers. At the same time, Tokusawa-en seeks to orient itself to the financially better-off customers; a dormitory bed here costs around 120 USD per person per night, and there are exclusive suite style rooms costing up to 500 USD per room per night; yet, there is so much demand among visitors that during most weekends in summer and autumn, the hut operates at its full capacity. The yearly total of lodgers is around 10,000; most lodgers belong to the advanced age group (above 50 years). Nearly half of the lodgers are casual hikers, while the other half are hikers/mountaineers/climbers. Peak demand coincides with summer vacation and autumnal foliage, and a large number of the lodgers belong to tour groups. The manager pointed out that international travelers were more likely to pay the premium price in order to stay in plush rooms. Regarding the natural aspects of the landscape, while he took pride in the surrounding vista, he complained that the forest has become 'overgrown' due to National Park restrictions on logging, and curiously, had the opinion that the agropastoral landscape of early 20th century was more 'natural'. He was also of the opinion that the landscape remains largely the same around the area, although specific aspects such as snowfall, flowering, and foliage timing have recently been undergoing yearly fluctuations.

The Tokusawa-en Mountain Hut also stands out for its large number of female staff (20 of 23 staff are women), and one of the staff pointed out that they consciously sought to deliver an image of the hut as a place for relaxation, tasty cuisine, and the warmth of hospitality. It remains to be pointed out that those aspects are still frequently associated with women in Japanese lodging facilities. The same respondent pointed out that she enjoys the vibe when customers spend time drinking beer and talking amongst themselves, although male customers at times tend to get a little too loud. She took pride that the mountain hut enjoys high popularity among female hikers (nearly 70% of overnight stayers are women). However, she also acknowledged that most hikers are unaware of the fine details of the landscape, and she did not think many are aware of its geology.

In contrast to Tokusawa-en, the Kamonji-goya is a no-frills facility. It is also one of the oldest huts, but lacks provisions of luxury and is usually used by older hikers who know the place well. Its first owner, Kamonji Kamijo, also became the first renowned guide for the Japan Alps, when he escorted the British missionary Walter Weston over a hundred years ago. Weston's travelogues in the area were key for introducing the North Japan Alps to the outside world; he is also often credited for coining the name 'Japan Alps'. At Kamonji-goya, only around 1000 people stay throughout the year, and the owner, a woman in her 70s, pointed out that it is one of the simplest huts in terms of facilities, but it is perhaps closest to what mountain huts looked like in Japan before the rapid economic development in the latter half of the 20th century. As she has been in the area for most of her life, she keenly perceived the changes in the landscape, and pointed out that the riverbed has risen by several meters in the last few years due to in-channel deposition of gravel in the Azusa River. She had also seen as many as 10 tunnels being opened up during her 50 year long association with the place, and made the interesting observation that during postwar development, the focus was on making the destination comfortable

and accessible for urban tourists, which resulted in Kamikochi becoming 'too easy to visit', leading to congestion, pollution of waterways, and alteration of the landscape. One of her other interesting remarks was that most people tend to go to places that are already well-known, and only a few are interested in off-the-beaten track experiences. Thus, even while she was eager to welcome visitors, she held a different view of the landscape, and after witnessing some of the longer term changes (on a human timescale), she did not interpret the place as remaining unchanging or pristine.

Category II: National Park Management: The National Park and visitor center staff, who form the formal management structure of the place, echoed the theme of the beautiful valley. Some key points that came out were visitor behavior, rules, no-littering, appropriate behavior. They sought to highlight problems such as visitor feeding of wild monkeys in the area, which in turn makes the monkeys more docile and dependent on offerings, and increases the chances of encounters with people at the same time. Typically, their vision of the landscape revolved around the concept of a beautiful playground that they consistently sought to keep open to as many people as possible, even though they voiced concern that visitation related problems are driving changes in the local wildlife. As it is obvious, there is some dichotomy in this vision for Kamikochi. In addition, National Park visitor centers also apparently highlight the visual beauty of the place along the lines of eternal and pure, perhaps in order to appeal to visitor image of Kamikochi.

Category III. Conservation scientists: On the other hand, scientists who worked on the compilation of the Natural History in the Kamikochi Valley tended to disagree in clear terms with the observation that the Kamikochi landscape is natural. One of them mentioned the significant impact of river engineering on vegetation species such as S. arbutifolia, which in turn influenced vegetation succession on gravel bars and altered riparian forest composition over the long term. Two of the respondents had also monitored the river morphology and riparian vegetation for well over a decade, and they emphasized the point that natural disturbance regimes are the driving factor for landscape composition and renewal, and it was anthropogenic alteration of such disturbance regimes, based on a static view of the riverscape frozen in time, that was responsible for the loss of spatial heterogeneity. Although they were aware of the potential effect of climate change on vegetation and other biophysical features, they feared that the ongoing homogenization of the riverbed into a single dominant channel (Figure 4) and the loss of fine-scale mosaic in the active riverbed was a more pressing threat.

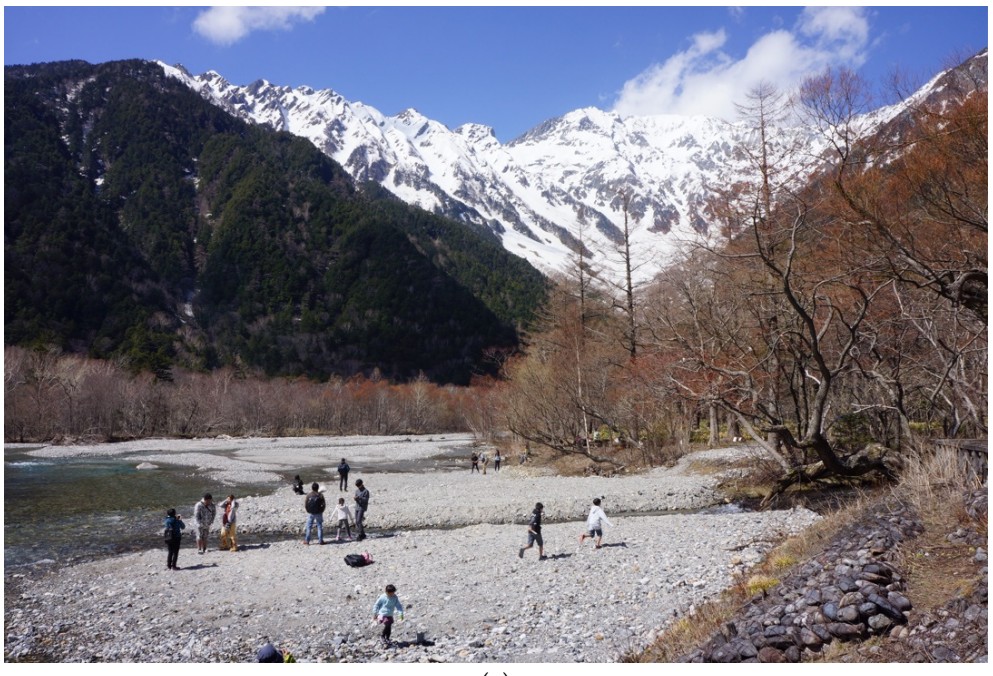

(**a**)

**Figure 4.** *Cont.*

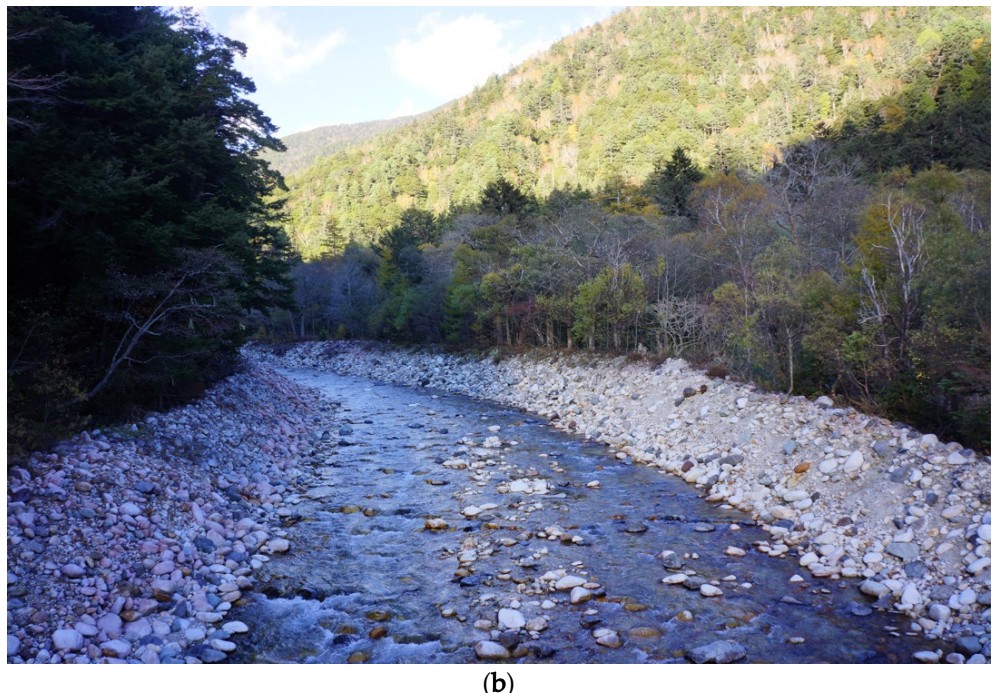

(b)

**Figure 4.** Subtle or clear anthropogenic modifications of the environment are widespread in Kamikochi. (**a**) (above) Tourists enjoying a sunny day on the gravelly bed of the Azusa River in Kamikochi; note the artificial enforcement of the bank in the foreground. (**b**) (below) The relatively straight single thread channel is the result of flow modification and embankment engineering. Photos provided by the author.

Personal observations: During several field trips in the region, it became clear that the landscape is both thoroughly constrained by human design and retains a powerful dynamic potential. Although the Azusa River may appear natural at Kamikochi, it is hardly a natural river, as cobbled embankments stretch all the way to the vicinity of its headwaters. Heavy machinery is present in the area throughout the year, and occasionally earth moving machines can be seen operating inside the river channel. This author witnessed in-channel gravel mining, boulder rearrangement, and construction or expansion of new trails (some of them are necessitated by snowmelt or landslide induced damage to existing trails). Yet whenever the river gets a chance, it reclaims its territory, as frequent bank erosion, hollowing out of soil from under the trails, and rock slides along tributary valleys demonstrate (Figure 5). During the early spring season every year, a temporary trail is opened along a portion of the riverbed, as the original narrow trail on the embankment is vulnerable to sudden rockfall and snowmelt induced mass movement. While there are guided tours in the area, most focus on a narrow view of explaining the biota (especially flowering plants that are visually attractive and popular among visitors) and tour guides typically do not venture into topics such as geology and recent anthropogenic changes in the river morphology. Visitors are also typically content to see Kamikochi as an ideal retreat from the hustle and bustle of urban life, and are seemingly satisfied with the stories of its serenity and beauty. This visitor inclination towards relaxation perhaps reinforces the epitomization of Kamikochi as a serene landscape, and fosters inadequate information about recent anthropogenic turmoil to geomorphological processes in that area, although some younger hikers seemed to be at least partly aware of this problem.

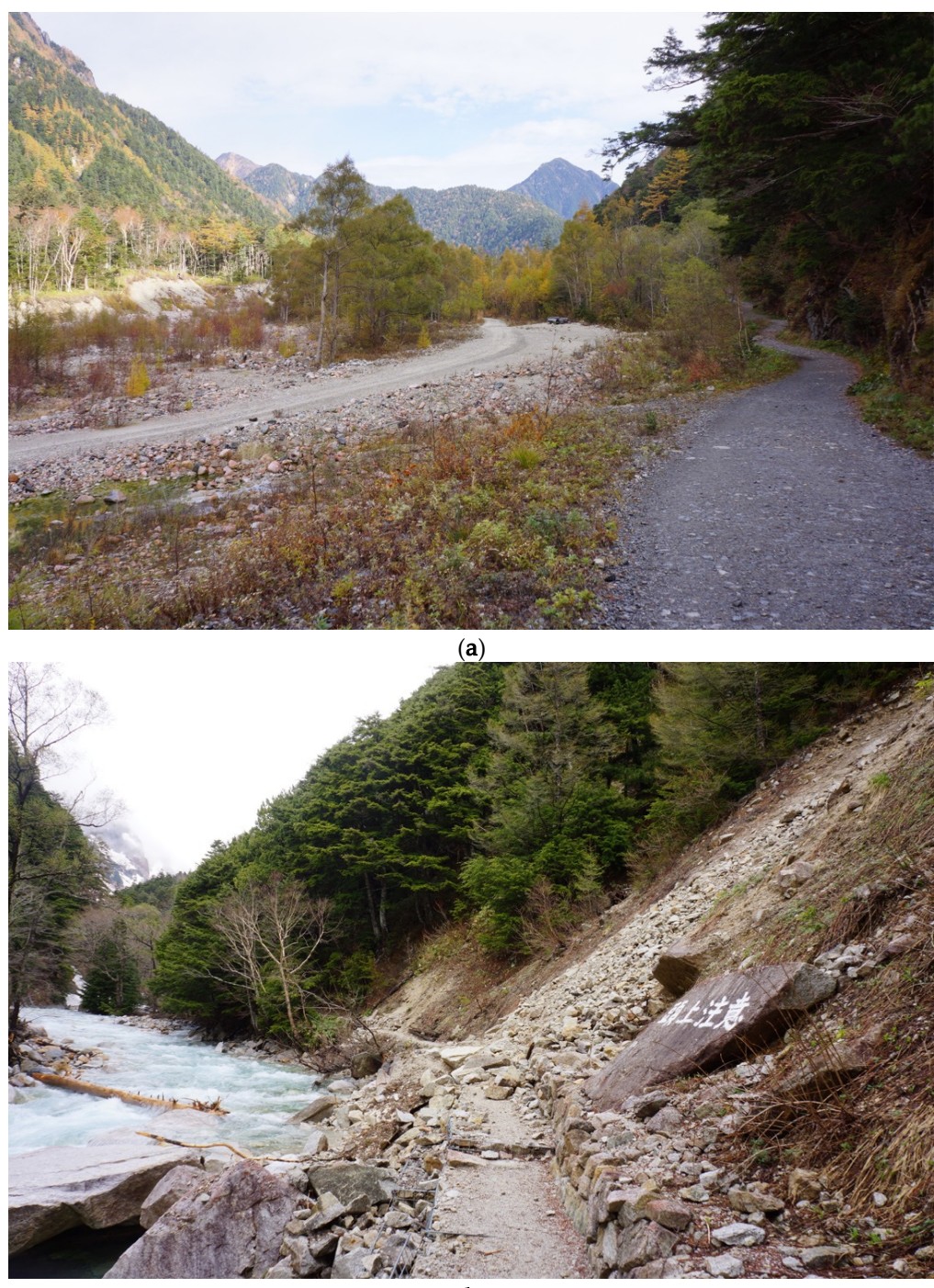

**Figure 5.** Human modification of the landscape and nature's forces to reclaim their territory working side by side. (**a**) (above) The main trail near Tokusawa area (on the right) and a supplemnatry trail (center) opened up to avoid spring snowmelt and landslides from above. (**b**) (below) Rockfall on the trail upstream: such dynamic landscape properties are essential for active riverbed formation. Photos provided by the author.

*4.3. Results of the Questionnaire Survey*

The questionnaire survey was administered during the second year of the project. It was aimed at gauging visitor characteristics, preferences, and consciousness about the Kamikochi landscape. Among the 80 valid responses, 43 were by women and 36 were by men, and 1 respondent replied 'other' as gender. Respondents mostly belonged to the advanced age-group: 33.8% were from the 50–60 years

old age-group and 23.8% belonged to the 60–70 years old age group. The results of the questionnaire are described below with graphical explanations:

As seen from the data in Figure 6, most of the respondents were familiar with the Kamikochi landscape, 51.3% replied that they visited the site between 2 to 4 times and 41.3% replied that they visited the place more than 5 times. In contrast, only 7.5% were first time visitors. A majority of the visitors (53.8%) were familiar with the National Park visitor center, with 36.3% replying that they had visited the facility multiple times.

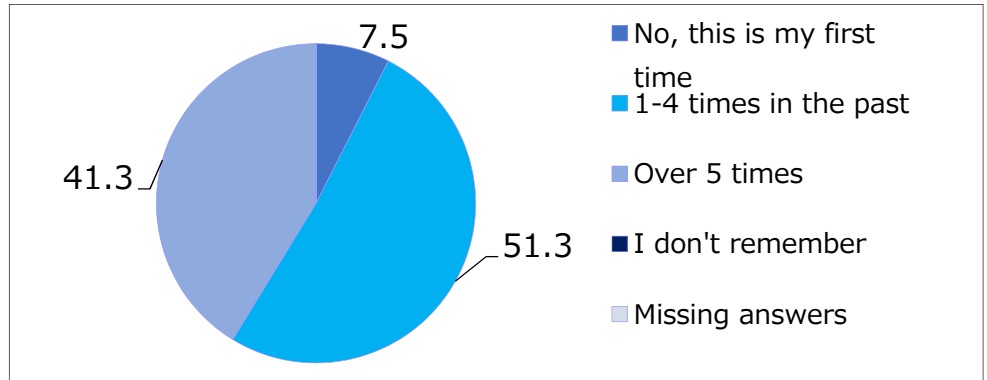

**Figure 6.** Answers for the question: Do you have previous experience of visiting this place (or surroundings)?

Regarding the type of activity they engaged in, Figure 7 shows that 60% of respondents were either part of a group or a family; and group tour or family travel were dominant objectives. A total of 26.3% respondents were solo hikers/climbers, and only 2.5% identified themselves as nature observers (Figure 7). The overwhelming majority of visitors (98.8%) stayed one night or more, and as many as 40% stayed for over 3 nights. Lodging is not cheap in the mountain hut where the survey was administered, this leads to two conjectures: (i) that most of the visitors are financially well-off and (ii) due to their advanced age, they preferred a slow mode of travel. However, when asked whether they were doing a circuit or traverse of the region, only 12.5% replied in the positive, indicating that most visitors remained within the valley and did not travel widely across the region.

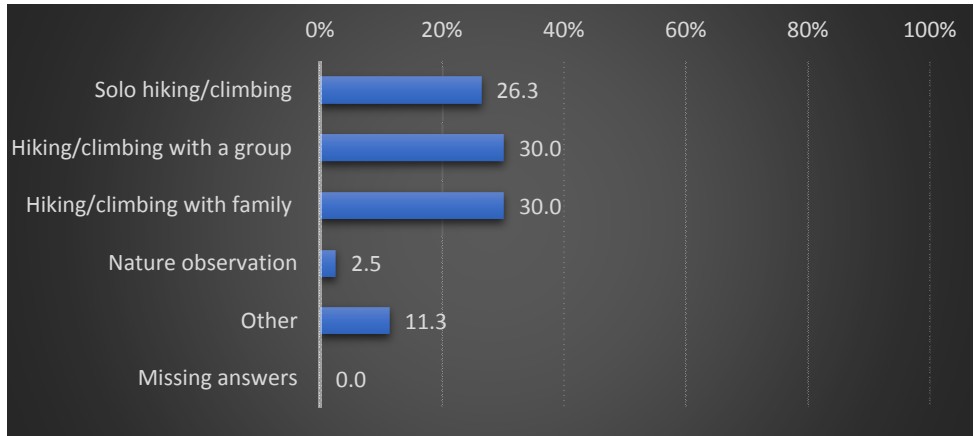

**Figure 7.** Types of visitors identified from the question: What is your objective for visiting the National Park this time?

When asked what aspects of the landscape they had most interest in, the category 'view from mountains and/or photography' was the overwhelming favorite with 90% response rate, while the category of 'mountain (peak)' was also chosen by 60%. Mountain vegetation came a close third with a 55% response rate. As multiple answers were possible, there is considerable overlap of preferences here,

but when contrasted with the 35% selection of the category 'human aspects/mountain hut culture', it is clear that the biophysical attributes of the landscape enjoy clear popularity among visitors (Figure 8).

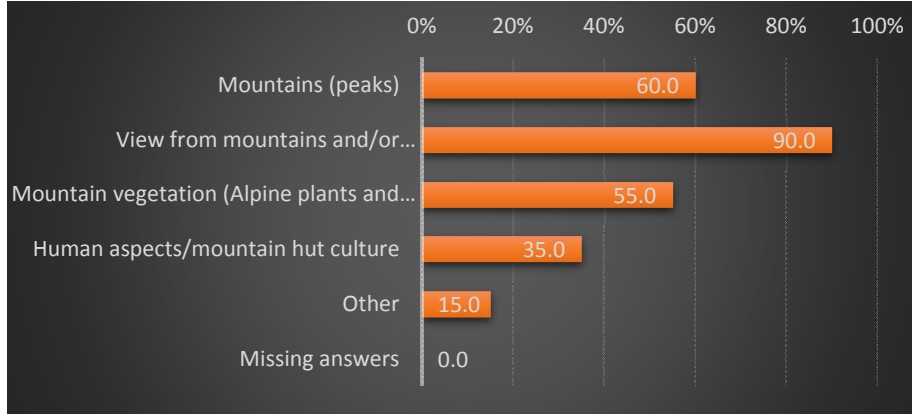

**Figure 8.** Visitor attractions revealed from the question: What is the most attractive feature for you in this place (and surroundings)? (Multiple answers were possible).

Yet, as Figure 9 depicts, despite their familiarity with the landscape and prior visits to the National Park Visitor Center, a majority of respondents indicated that they did not understand much about the biophysical foundations of the environment such as its geology, geomorphology, and ecology; an additional 12.5% of respondents replied that they had nearly no knowledge about those aspects.

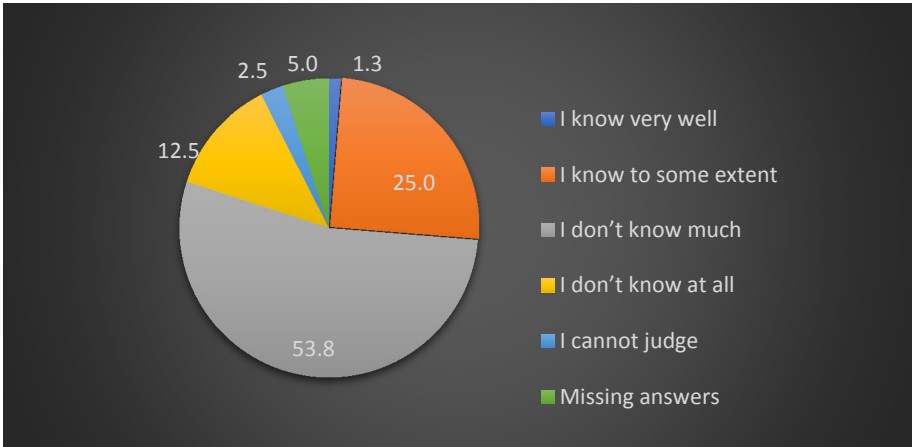

**Figure 9.** Visitor self appraisal of environmental knowledge from the question: How much do you think you know about the geology, geomorphology, and ecology of North Japan Alps?

When asked if they were aware of 'changes' to the ecosystems of the North Japan Alps, 63.8% answered in the affirmative. However, when asked where they perceived the 'change' to be occurring, 36.3% of the respondents could not provide an answer, while 33.8% identified 'climate', 35% identified 'animals', and 23.8% identified 'plants.' Multiple selections were possible for this question, so there is some overlap among the change aspects identified by the respondents. Interestingly, changes in the river morphology were identified by only 21.3% of visitors, indicating that visitor awareness of extensive human modification of the river course and fluvial properties remains low (Figure 10).

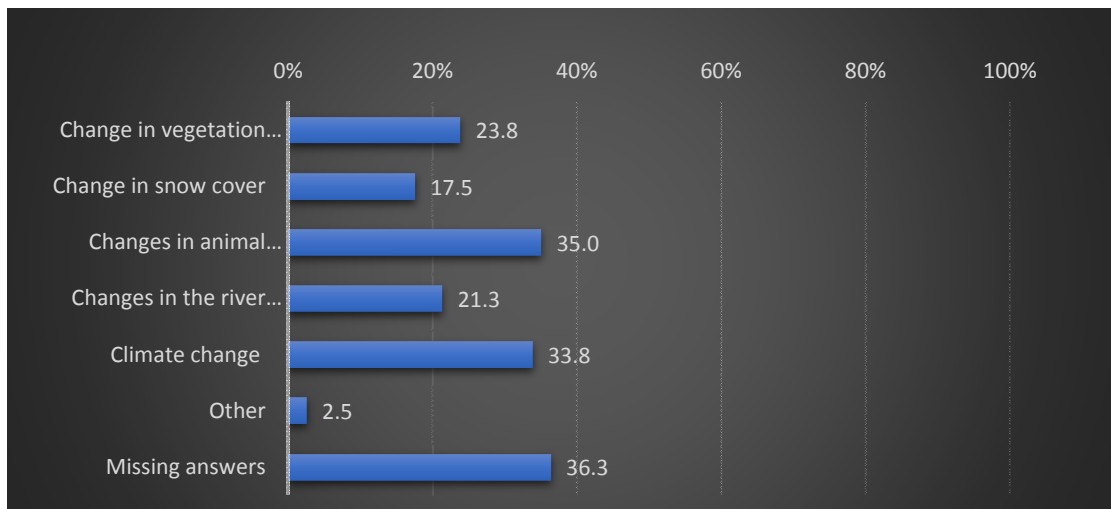

**Figure 10.** Visitor perceptions of changes in the environment (multiple answers were possible).

When further asked about what factors they saw as being responsible for the changes, only 17.5% identified overuse of natural resources, while 30% chose 'natural hazards' as the cause. Although 45% identified global environmental change as a possible factor (again, multiple answers were possible for this item); the anomaly in the responses is clear (Figure 11). These aspects are further explained in the 'Discussion' section below. Finally, when asked if they would participate in any ecotour program that included explanations on the geological, geomorphological, and ecological aspects of the area, the majority (53.8%) replied in the affirmative, indicating a clear demand for such information among visitors.

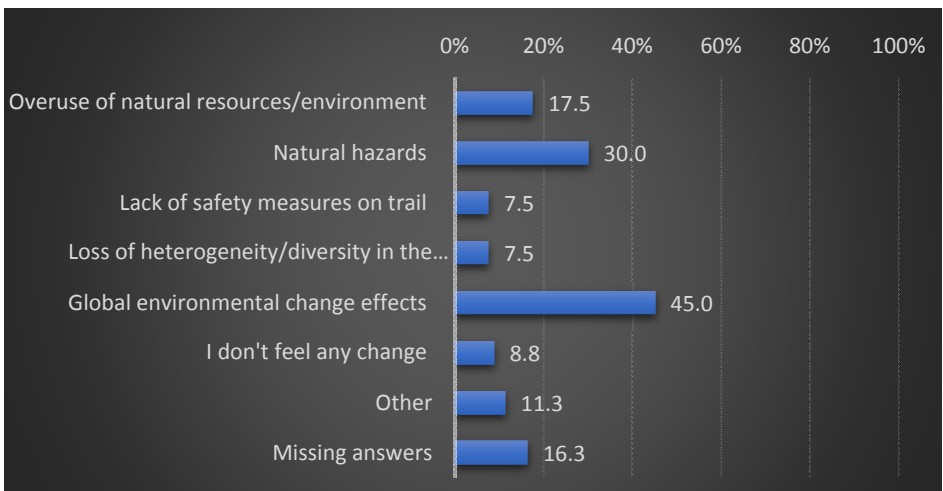

**Figure 11.** Visitor perceptions of causes behind changes in the environment.

## 5. Discussion

As the interview and survey data reveal, there is an apparently high degree of appreciation regarding the biophysical aspects of the mountain landscape. However, there are also clear differences between types of stakeholders regarding how those aspects are perceived, interpreted, or valued. While conservation scientists emphasized the dynamic, heterogeneous, and at times unruly landscape characteristics, tourism service providers generally portrayed the place as having a fixed characteristic, which is based on the ideal of a scenic retreat for urbanites. This view of the landscape is also bolstered by the National Park narrative of a beautiful Kamikochi. Tourism service providers were aware of some changes to the environment, but they were mainly sensitive to seasonal or decadal changes,

due to the fact that aspects such as flowering and foliage timings form the bases of visitor demand. They typically did not show high awareness of anthropogenic changes to the geomorphological or biological aspects of the area, or of changes that are not readily visible, although some respondents who were associated with the place for multiple decades did voice their apprehension towards ongoing man-made changes near landscape features such as widespread tunnel and road construction. The National Park management sought to highlight problems such as feeding wild animals, off-trail walking that damaged fragile plants, and littering of trash; but did not exhibit a clear stance on the extensive modification of the hydrological properties of the Azusa River. As identified by Iwata (2016) [21] and Iwata and Yamamoto (2016) [32], the construction of tributary streams for controlling gravel flow into the primary channel, straightening of the main flow, and embankment fortification and road/trail expansion on the embankments all carry the negative impact of homogenizing the active riverscape. In addition, suppression of natural disturbance regimes is causing changes in vegetation such as for the *S. arbutifolia* colony, which also serves as an indicator for the vigor and integrity of the natural disturbance regimes.

Most visitors surveyed in the study were familiar with the place, as they had visited it multiple times, but at the same time their responses indicated that they had very little information about geological and geomorphological aspects of the landscape. They also did not get much information about ongoing human modification of the place, despite visiting National Park information centers multiple times. This suggests that there is an urgent need to provide information on long-term anthropogenic impacts on the landscape, as well as dynamic landscape mechanisms operating over geological time. There is clear solidarity among visitors with the visually attractive parts of the landscape, such as the mountain ridgeline and flowering plants. The majority of respondents identified several changes in the environment of the North Japan Alps in general, but most could not identify specific changes to the place they were in, and their interpretation of major challenges for the landscapes sometimes yielded anomalous answers to claims made by conservation scientists (such as the choice of 'hazards' as a major cause of change), which is probably explained by the lack of information on the part of visitors that was highlighted several times in their own responses. In addition, most visitors apparently did not explore the area widely and remained confined to specific lookouts.

With the backup of personal observations of this author, it can be argued that each of the positions represented by the interviewees is logical, and that the anomalies stem from the type of association the particular individual enjoys with the landscape, the length of that association, and his/her preferred vision of the landscape. Time emerged as a key factor behind respondents' perceptions of the landscape. Respondents below 50 years of age and those with less than 20 years of constant association perceived annual fluctuations keenly but were not always aware of changes over longer timescale. On the other hand, respondents who were associated with the place for longer time were aware of changes dating back further in the past, but only as far as their memory helped them. Visitors typically had a shallower knowledge of the landscape across time, and while they could identify broad-scale problems, they could not point out specific changes. Conservation scientists were the only group that had the grasp of changes operating over the longest timescale—i.e., geological time—and their view of the landscape as constantly oscillating due to episodic volcanism, uplift, glacial and river erosion, and the transportation of materials from the ridge to lowland remains vital for addressing the integrity of this dynamic place. There are indications that this perspective is currently missing from management priorities, and that there is an urgent need to incorporate it into the planning fold.

Extending the insights to the international context, it can be posited that tourism stakeholders, especially visitors, possess a high interest in visually appealing aspects of the environment and can be willing to contribute to conservation interests. As a case study in the Eastern Ore Mountains of Germany demonstrated, availability of nature-based experience and visually attractive landscapes are major pull factors for visitors who generally tend to show a willingness to pay for protecting those aspects [48]. Findings from the Kamikochi Valley positively correlate with the broad patterns of this study. However, it should also be kept in mind that there are differences in stakeholder attitudes and

perceptions depending on their social and cultural backgrounds, as revealed by a comparative study of visitors of different nationalities by Priego et al. (2008) [49], and planning inputs must be formulated upon careful deliberation of such characteristics.

It can also be pointed out that this case study represents a parallel to the US scenario involving the construction of the O'Shaughnessy Dam at Hetch Hetchy Valley in the early 19th century. In the case of Hetch Hetchy, the dam was eventually built and the picturesque valley was inundated by the reservoir, but that episode became instrumental for raising public awareness towards nature conservation and the institutionalization of the US National Park Service in 1916 [50].

In a broader context, dynamic physical processes operating over million year timescales and ecosystem responses to natural disturbance regimes engender heterogeneity and visual beauty of mountain landscapes—yet when packaged for tourism, only certain parts of that dynamic whole is valued and communicated, while the underlying structure of the landscape is subjected to continuous modification. Mountain landscapes that are easily accessible and popular such as Kamikochi are constrained by their developmental pathway that facilitates mass consumption. The resilience of such landscapes has probably declined over time as this case study suggests, and anthropogenic changes to their physical properties and processes make these places more vulnerable to shocks such as climate perturbations. So far, tourism development and landscape conservation have largely progressed on opposing trajectories, and this situation has led to tourism being a part and parcel of the wider human modification of earth processes in mountain environments. However, as the findings of this study also indicate, there is a coalescence on the value of the biophysical landscape among stakeholder types, and if tourism planners can work with conservation scientists under the fold of protected areas such as National Parks, tourism can possibly incentivize conservation of dynamic properties of such landscapes.

Future research: This study provided important management indicators such as the clear demand of geological, geomorphological, and ecological information on the part of visitors, the lack of information about ongoing anthropogenic modification of spatial heterogeneity and natural disturbance regimes, and the relatively simple nature of visitor interaction with the Kamikochi Valley. It will be pertinent to design management and visitor education programs based on these insights and measure their efficacy over time. In addition, further monitoring of environmental change in spatial and temporal dimensions will be needed.

## 6. Conclusions

This article provided an analysis of sustainable tourism challenges in a dynamic landscape through the case study of Kamikochi Valley of North Japan Alps. As one of the signature mountain destinations in Japan, the area is subjected to intense visitor pressure from spring to autumn. The intensity of visitation results in direct pressure on the landscape and wildlife, as well as in more subtle pressure in the form of ongoing infrastructure buildup and modification of key geomorphic processes. The active riverbed of the Azusa River encapsulates a complex range of processes such as past volcanism, ridge formation in the Quaternary, glacial erosion during recent glacial maxima, as well as Holocene deglaciation and high rates of mass movement/denudation. However, such processes are inadequately perceived in the planning mechanism, as well as by individual tourism service providers and visitors. The expansion of tourism has favored a static and risk averse approach to landscape management, which has resulted in obstruction or modification of key landscape level processes. While visually appealing aspects such as the ridgeline and flowering plants are keenly appreciated for their beauty, the fluctuating and at times unruly nature of the natural landscape itself does not enjoy enough attention from guiding tours or visitor information contents. The visitor survey revealed that while visitors are aware of issues such as climate change, they typically do not have an adequate understanding of geological and geomorphological properties of the place. As the survey also revealed a general demand of such information, it remains an urgent task to provide information on the dynamic landscape and its current vulnerability to visitors. Urgent measures are also required to ensure that the place is managed

with its natural change pathways and heterogeneity in mind. Finally, as mountain landscapes are highly dynamic and their evolution and resilience properties are highly location-specific, the overarching challenge for managing tourism in a sustainable manner remains in understanding, appreciating, and proactively conserving the biophysical mechanisms of such places.

**Funding:** This research was funded by the JAPAN SOCIETY FOR THE PROMOTION OF SCIENCE, Grant number 19K20567.

**Acknowledgments:** The author is indebted to Yuichi Aoki of Waseda University Research Institute of Manifesto for statistical representation of the survey data. The staff of Tokusawa-en are thanked for their cooperation in administering the survey. The author also wishes to thank the reviewers whose comments and suggestions made the work better.

**Conflicts of Interest:** The authors declare no conflict of interest. The funders had no role in the design of the study; in the collection, analyses, or interpretation of data; in the writing of the manuscript, or in the decision to publish the results.

## Appendix A

**Figure A1.** Questionnaire used in the study (Translated version).

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
