# Peer review of "Emerging Patterns of Mountain Tourism in a Dynamic Landscape: Insights from Kamikochi Valley in Japan"

_land, doi:10.3390/land9040103_

Round 1
Reviewer 1 Report
A very interesting and innovative paper on a subject of considerable topical interest. The point on lines 107-110 make an interesting comparison with the Hetch Hetchy controversy in the US at the end of the 19th century and the establishment of Yosemite and other parks, again a proposed dam to flood a valley. That case, and also some comparisons with changing human perceptions of mountain landscapes are found in the works of the Canadian Gordon Nelson, who has done a great deal of work in this area.
The paper is important for park managers and others as very often they are unaware of visitor and operator perceptions and rarely take these into account in the management of such regions.
Author Response
Sincere thanks for your positive review of my work.
Also, thank you for reminding that the Hetch Hetchy case remains an instructive example of mountain landscape-anthropos interactions. I have inserted a brief note in the Discussion section mentioning the Hetch Hetchy case. I do not have Gordon Nelson's works at hand, but I have cited Righters (2006) detailed work in that section.
PS: I have inserted a note thanking all reviewers for their comments/suggestions which made the work better.
Reviewer 2 Report
The paper is an interesting research on a two-year long research project at the Kamikochi Valley 26 of North Japan Alps.
The research is very useful and the paper can be accepted with moderate revisions.
Check carefully for typos the text, especially in the Results.
Below a few comments/suggestions for the improvement of the paper
Add the ‘long-term perspective’ as ‘history of the land’ in the Introduction -
In the Introduction it is correctly reported that “relatively little literature is available on anthropogenic alteration of physical processes that engender landscape diversity of mountain destinations”: the paper examines many factors, and I suggest to add also the long-term perspective of anthropogenic influence or alteration on the environment, including mountains, because when deciding how and how much changes are possible in mountain destinations (aiming at “recognizing the integrity of key physical processes and heterogeneity of the landscape “) one must take into account the actual environment-human history of the region. This history can be older than the 20th century, thus revealing the ancient vocation of the Valley and the way its development may be in the direction of a sustainable tourism. Only in this way one can decide if the current landscape is – partly or totally - ‘natural’ or ‘anthropogenically-shaped’.
Move paragraph 3.1-
It is correct that the ‘Natural History in the Kamikochi Valley’ is considered a method of study but it is not so clear why it is reported in the Results: the section of results should report only the outcomes from the questionnaires (also about this topic) and not the description of the content of that research (lines 169-192 may be moved to the Introduction, in a special section – and then other parts are more proper argument of the Discussion).
Move paragraph 3.3. before the 3.2 (?) –
The paragraph 3.3 is the output of questionnaires and therefore seems to be the presentation of results more than the other parts. As it is clear that results must be introduced before any discussion-inferences or conclusion, probably one should introduce the set of interviews etc in a more systematic way in the methods.
Clarify the number of interviews (?) -
This is the same observation that again ask for a more clear correspondence between methods and results sections. It is a pity that, despite the high number of questionnaires correctly distributed, only the number of data from 80 samples were collected. But, in
line 217 it is reported that 7 interviews are avialable. Please clarify in the Methods.
Author Response
Thank you for the positive feedback. I have addressed all of your concerns and made changes where appropriate in the revised version, please see highlighted parts in the revised manuscript as well as the attachment to reviewer comments where specific responses are displayed on a table format.
In addition a tracked version of the manuscript has also been uploaded that can be used to confirm all changes including wordings, typos, and punctuation.
PS: I have inserted a note thanking all reviewers for their comments/suggestions which made the work better.
Reviewer Comment |
Author Response |
The research is very useful and the paper can be accepted with moderate revisions. Check carefully for typos the text, especially in the Results. |
Thank you for the positive feedback.
I have done this in the revised version now.
|
Add the ‘long-term perspective’ as ‘history of the land’ in the Introduction - In the Introduction it is correctly reported that “relatively little literature is available on anthropogenic alteration of physical processes that engender landscape diversity of mountain destinations”: the paper examines many factors, and I suggest to add also the long-term perspective of anthropogenic influence or alteration on the environment, including mountains, because when deciding how and how much changes are possible in mountain destinations (aiming at “recognizing the integrity of key physical processes and heterogeneity of the landscape “) one must take into account the actual environment-human history of the region. This history can be older than the 20th century, thus revealing the ancient vocation of the Valley and the way its development may be in the direction of a sustainable tourism. Only in this way one can decide if the current landscape is – partly or totally - ‘natural’ or ‘anthropogenically-shaped’. |
Agreed, long-standing human-landscape interaction in the mountain context is an important driver in landscape evolution. I have added this part with some appropriate references in the introduction, and I have also added more information on this angle when discussing the landscape evolution of the Kamikochi Valley. However as my research shows, all human-landscape relationships are relatively recent when compared to the natural landscape processes that operate over geological time. My central message therefore remains that the integrity of those biophysical processes should be addressed more proactively.
|
Move paragraph 3.1- It is correct that the ‘Natural History in the Kamikochi Valley’ is considered a method of study but it is not so clear why it is reported in the Results: the section of results should report only the outcomes from the questionnaires (also about this topic) and not the description of the content of that research (lines 169-192 may be moved to the Introduction, in a special section – and then other parts are more proper argument of the Discussion). Move paragraph 3.3. before the 3.2 (?) – The paragraph 3.3 is the output of questionnaires and therefore seems to be the presentation of results more than the other parts. As it is clear that results must be introduced before any discussion-inferences or conclusion, probably one should introduce the set of interviews etc in a more systematic way in the methods. |
I have added an explanation in the Results section: The findings are reported in three sections following the actual chronological sequence of the research project: i) content analysis from “Natural History in the Kamikochi Valley”(long-term monitoring of the place) conducted April 2017-March 2018 (ii) 7 in-depth interviews and personal observations of the author conducted April 2017-March 2018, (iii) structured questionnaire survey (conducted June 2019 and October 2019). The sequencing is therefore appropriate and as contents analysis is a standard form of analysis in qualitative research its findings are described in the ‘Results’ section. See added text in revised manuscript.
Kindly refer to the response above. Also, the section numberings were inaccurate, the correct form would be 4.1, 4.1…and not 3.1, 3.2…I have corrected those now. I have also added an explanation in the ‘Materials and methods’ section to make things clear. |
Clarify the number of interviews (?) - This is the same observation that again ask for a more clear correspondence between methods and results sections. It is a pity that, despite the high number of questionnaires correctly distributed, only the number of data from 80 samples were collected. But, in line 217 it is reported that 7 interviews are avialable. Please clarify in the Methods.
|
There were 7 open ended interviews that were particularly useful during the first phase of the research in addition to the 80 questionnaire samples. I have added an explanation to make this clear in the revised manuscript. |

Reviewer 3 Report
See attached file

Author Response
Thank you for reviewing my paper. I thank you for the comments and I have addressed them to the extent possible. However it appeared to me that most of the comments/suggestions were either from (i) a lack of understanding of what qualitative research, description, and reporting results entail and (ii) a matter of personal preference regarding writing style etc.
Specifically the comments seem to converge on the position of omitting description/anecdotes and prioritizing numerical analysis (with the suggestion of a chi square test on the data). This is not possible because that would result in compromising the integrity of my research, its design, and analysis framework. This is an inappropriate suggestion because this work does not intend to demonstrate results of statistical hypotheses testing (i.e. it does not set out to prove statistical correlation in order to explain causality), and it will be wrong to conduct chi-square tests on this type of qualitative data. Furthermore statistical correlations such as Chi-square testing are not omnipotent solutions for everything, and are hardly considered appropriate for reflective, nuanced analysis of qualitative research (see point below).
I respectfully differ on those aspects and provide my responses below. Also kindly refer to the attached MS Word File.
In addition a tracked version of the manuscript has also been uploaded that can be used to confirm all changes including wordings, typos, and punctuation.
PS: I have inserted a note thanking all reviewers for their comments/suggestions which made the work better.
Reviewer comment |
Author response |
The writing style is extremely wordy and conversational. I believe that the paper could be reduced to half its length without difficulty, mostly be taking out a lot of unnecessary (conversational) words. The very long paragraphs are difficult to read and should be broken up – apply the rule “one paragraph, one point”.
|
This is an opinion on writing style. Please take note that thick description, contextualizing information, and providing reflectivity/nuance are all strengths of qualitative research. Qualitative analysis does not need to conform to the accounting-like efficiency of one paragraph-one point, rationing of words, and the singular mindset of prioritizing numerical analysis above all else. You are welcome to refer to any of the 8 references on qualitative research provided in my paper; or, the excellent sourcebook on qualitative analysis by Michael Patton (cited in paper) or, Denzin & Lincoln’s SAGE Handbook of Qualitative Research (any edition) for better understanding of this type of research. These are sources that are followed by a very large number of qualitative researchers as well as being taught in many graduate level courses in qualitative research throughout the world, so they are quite reliable! One of my own recent works in Tourism Geographies journal extended those arguments https://doi.org/10.1080/14616688.2020.1713875 You are welcome to refer to that too. |
The analysis is very simple, being restricted to pie graphs and some rather unusual histograms. The pie graphs should be eliminated – the data can be incorporated into the text or a table should be prepared. I also doubt that the histograms are really needed. Whatever the decision here, there should be a complete review of how the data are summarised and presented because the current presentation is unwieldy, not very informative, and not typical for a scientific journal. I believe that some chi square testing would be possible on some of the claims made, using the data that are presented.
|
I believe the graphical representations of the data are useful, and hence I have chosen to retain them. Also see my general comment above for the inappropriateness of the Chi-square testing for this type of data. |
The study is descriptive and some of the reporting is anecdotal (from in-depth interviews) and qualitative. While those aspects are ok, they need to be reduced in length
|
Description, anecdotal evidence, and reflectivity are strengths of qualitative research. The aims, design, and framework of the study are clearly explained in the materials and Method section, please refer to that information. I have merely followed a very standard pattern of research and reporting in this article. |
The results section begins with material that is not results. It is further introduction and description of the location. The results really begins at Section 3.2, although even here this section begins with introductory comments and methods and contains a lot of commentary that is not results. For example (line 230):
|
The results begin with material that is ‘Result’! of content analysis, and follows up with Results from open-ended interviews/observations, and finally the questionnaire. It is not that simply because the questionnaire gives some numerical data, only that forms the result of the study. Once again, there is a misunderstanding of qualitative research here. For ease of understanding I have added text in the Results section to clearly explain the framework of results, but also refer to my responses above. |
None of this is really results, as it is a description of a mountain hut in the study area, and the phrasing in yellow suggests that the information is imprecise and not based on data. The typos marked in blue are typical – there are a large number of typos in the manuscript needing attention.
|
Elaborating on the context and explaining the natural setting of research is a vital part of qualitative studies. Refer to explanations in the Materials and Method section. The information you marked in yellow are direct information from respondents. For example when the respondent says ‘…nearly half of my boarders are casual hikers…’ the researcher is obligated to report that statement factually (i.e. retain the message of ‘nearly’ in reporting). There is no use of seeking numerical accuracy here, and simply not reporting the information or assigning an arbitrary value that was not reported by the respondent will be less than accurate. So, while less than numerically specific information is still important information (at least in qualitative research) but modification of data just to provide an appearance of accounting-style efficiency in writing is not appropriate. Thanks for noticing the typos, I have rectified wherever I found them and will be happy to correct if anything comes up during typesetting. |
RE: the suggestions of styling, reporting etc. |
These are matters of personal opinion. I have followed a standard approach of reporting in the paper and have been consistent during research. Once again I seek to emphasize the following point: this work is based on robust, time consuming planning and method, and the results are delivered from this sound academic footing, and make a clear contribution to the field; a fact that has been recognized by all other reviewers. |

Reviewer 4 Report
The article is in general well written, shows evidence of a deep research, and is of potential interest to a broad international audience. However, fine tunning, especially related to structuring and writing up the research, are required before publication.
In more details, the final paragraph of the introduction repeats the abstract, presenting a summary of findings and their implications. They do not belong to the introduction, but to the abstract. The introduction should end with the sentence concluded on line 56.
The discussions would benefit upon expanding them to include a comparison with other similar studies carried out in other countries, and also underlining the specificity of the Japanese findings. For example, other studies indicated that there are differences between the perception and use of nature - see Priego, C.; Breuste, J.; Rojas, J. Perception and Value of Nature in Urban Landscapes: A Comparative Analysis of Cities in Germany, Chile and Spain. Landscape Online 2008, 7, 1-22 (https://www.landscape-online.org/index.php/lo/article/view/LO.200807).
Last but not least, the English language needs polishing, by the means of assistance form a native speaker. While the article can be understood, the pharsing and choice of words are often ackward, and hinder the understanding of the article.
Author Response
Thank you for your positive feedback on my work. I have incorporated/addressed all your suggestions in the revised manuscript. Please see the added text (highlighted) in the revised manuscript as well as the attached Response to reviewer MS Word file that provides specific responses on a MS Word table format.
In addition a tracked version of the manuscript has also been uploaded that can be used to confirm all changes including wordings, typos, and punctuation.
PS: I have inserted a note thanking all reviewers for their comments/suggestions which made the work better.
Reviewer Comment |
Author Response |
The article is in general well written, shows evidence of a deep research, and is of potential interest to a broad international audience. However, fine tunning, especially related to structuring and writing up the research, are required before publication.
|
Thank you for your positive feedback; I have addressed your concerns in the revised manuscript. Please see highlighted sections for added information/text. |
In more details, the final paragraph of the introduction repeats the abstract, presenting a summary of findings and their implications. They do not belong to the introduction, but to the abstract. The introduction should end with the sentence concluded on line 56.
|
Yes, this read repetitive, I agree with your observation. I have excised this fragment following your suggestion. |
The discussions would benefit upon expanding them to include a comparison with other similar studies carried out in other countries, and also underlining the specificity of the Japanese findings. For example, other studies indicated that there are differences between the perception and use of nature - see Priego, C.; Breuste, J.; Rojas, J. Perception and Value of Nature in Urban Landscapes: A Comparative Analysis of Cities in Germany, Chile and Spain. Landscape Online 2008, 7, 1-22 (https://www.landscape-online.org/index.php/lo/article/view/LO.200807).
|
Thank you for this suggestion. I have now expanded this section incorporating the literature you suggested, another quite pertinent case study from Eastern Ore mountains of Germany, and with a reference to the Hetch Hetchy controversy in the US. |
Last but not least, the English language needs polishing, by the means of assistance form a native speaker. While the article can be understood, the pharsing and choice of words are often ackward, and hinder the understanding of the article.
|
This has been done. |

Round 2
Reviewer 3 Report
This has been barely changed, so my original comments remain the same. It still contains the long-winded writing, the many typos, and the poor scientific presentation of data, that I have already responded to.
For example, from the results section 4.1:
As noted above, a detailed account of the natural environment of the Kamikochi Valley was
recently compiled by a group of local conservation scientists who studied the place for nearly three
decades [43]. This complitaion is especially valuable as it provides insights from long-term
monitoring of the environment—a rarity in environmental research literature in Japan. The principal
characteritcs of this dynamic landscape as documented in this work and pertinent information from
more general literature are summed up below as main results of conetnt analysis:
The above paragraph includes introduction, methods, and discussion, and even includes a reference. None of it is results.
The author has also chosen to attack the reviewer rather than make a genuine attempt to revise the paper following at least some of the suggestions. I note that this paper presents some data quantitatively (although poorly), so arguing that it is a qualitative paper and so the problems relate to my lack of understanding of qualitative methodology rather than to the author's poor writing are at the minimum, contradictory. It is not relevant, but I have been involved in considerable qualitative research and have faced the challenges of creating a concise overview of enormous amounts of detail.
If the author is directly quoting comments from interviews, then that should be clear.
The comment about chi square was supposed to be supportive - designed to improve aspects of a presentation that is already quantitative. It was not a demand for a quantitative analysis. On a separate point, no science methodology textbook will recommend using pie charts. The data in them can be presented much more concisely. They do not belong in a scientific presentation, whether qualitative or quantitative.